# Technical note: Analytical protocols and performance for apatite and zircon (U-Th)/He analysis on quadrupole and magnetic sector mass spectrometer systems between 2007 and 2020

Cécile Gautheron[1], Rosella Pinna-Jamme[1], Alexis Derycke[1], Floriane Ahadi[1], Caroline Sanchez[1], Frédéric Haurine[1], Gael Monvoisin[1], Damien Barbosa[1], Guillaume Delpech[1], Joseph Maltese[2], Philippe Sarda[1], Laurent Tassan-Got[2]

[1]Université Paris-Saclay, CNRS, GEOPS, 91405, Orsay, France
[2]Université Paris-Saclay, CNRS/IN2P3, IJCLab, 91405 Orsay, France

*Correspondence to*: Cécile Gautheron (cecile.gautheron@universite-paris-saclay.fr)

**Abstract.** Apatite and zircon (U-Th)/He thermochronological data are obtained through a combination of crystal selection, He content measurement by heating crystal and analysis using noble gas mass spectrometry, and measurement of U, Th and Sm contents by crystal dissolution and solution analysis using inductively coupled plasma mass spectrometry (ICP-MS). This contribution documents the methods for helium thermochronology used at the GEOPS laboratory, University Paris Saclay, between 2007 and the present, that allow apatite and zircon (U-Th)/He data to be obtained with precision. More specifically, we show that the He content can be determined with a high precision using a calibration of the He sensitivity based on the Durango apatite and its use also appears crucial to check for He, U-Th-Sm analytical problems. The Durango apatite used as a standard is therefore a suitable mineral to perform precise He calibration, and yields (U-Th)/He ages of 31.1±1.4 Ma with an analytical error of less than 5% (1 σ). The (U-Th)/He ages for the Fish Canyon Tuff zircon standard yields a dispersion of about 9% (1 σ), with mean age of 27.0±2.6 Ma comparable to other laboratories. For the long-term quality control of the (U-Th)/He data, attention is paid to evaluate the drift of He sensitivity and blanks through time as well as that of (U-Th)/He ages and Th/U ratios (with Sm/Th when possible), all relying on the use of Durango apatite and Fish Canyon Tuff zircon as standards.

## 1 Introduction

Apatite and zircon (U-Th)/He thermochronology (AHe and ZHe respectively) is now a mainstream tool to reconstruct the Earth evolution through history of cooling and exhumation over the first dozen of kilometers (e.g. Farley, 2000; Farley, 2002; Gautheron and Zeitler, 2020; Reiners, 2005; Reiners and Brandon, 2006). The geological implications of these data rely on the precision of measurements of He, U, Th and Sm contents of apatite and zircon crystals, by: (i) crystal picking and the

accuracy of the alpha ejection correction though crystal shape measurement; (ii) non-destructive He degassing and abundance determination by mass spectrometry; (iii) U, Th and Sm analysis after crystal dissolution and solution analysis by ICP-MS, and also on knowledge of He diffusion. Different contributions have already presented parts of the analytical protocols, for example, for crystal dimensions measurements (e.g. Cooperdock et al., 2019; Glotzbach et al., 2019; Herman et al., 2007), He degassing using laser beam (e.g. Foeken et al., 2006; House et al., 2000), dissolution and analysis of U, Th, Sm, Ca or Zr (e.g.

Evans et al., 2005; Guenthner et al., 2016; Reiners and Nicolescu, 2007) or improvements of the noble gas analysis by magnetic sector mass spectrometry (Burnard and Farley, 2000). In addition, several contributions have also been dedicated to the determination of He diffusion in apatite and zircon (e.g., Farley, 2000; Flowers et al., 2009; Gautheron et al. 2009; Gautheron et al., 2020; Gerin et al., 2017; Guenthner et al., 2013; Goldsmith et al., 2020; Shuster et al., 2006).

       In this contribution, we present all the methodologies developed and used in the GEOPS laboratory, University Paris

Saclay, for more than a decade, focusing on the He degassing process and He content analysis, acid digestion, U and Th (Sm and Ca for apatite) analysis by ICP-MS and data reduction, that all lead to (U-Th)/He thermochronological data. The detailed protocols are one among many others, and this technical note as no other purpose than to present them. Specifically, analytical details are given on two built-in-house noble gas extraction-purification lines, one coupled to a quadrupole mass spectrometer and the other to a magnetic sector mass spectrometer. An efficient method of He content calibration using fragments of the

Durango apatite standard is presented. Finally, (U-Th)/He data obtained for 6 to 8 months in the laboratory for Durango apatite and Fish Canyon Tuff (FCT) zircon are presented. The wealth of data, i.e., 272 apatite and 57 zircon measurements, does allow to gain new insights about data acquisition, analytical difficulties, and the reproducibility of He, U, Th (Sm for apatite) contents and (U-Th)/He data.

## 2 Methods

**2.1 Apatite and zircon (U-Th)/He thermochronological methods**

       The (U-Th)/He thermochronological method is based on the accumulation within the crystal structure of $^4$He atoms produced by the decay of $^{238}$U, $^{235}$U, $^{232}$Th and $^{147}$Sm to alpha particles that become $^4$He atoms, and it exploits the temperature-dependent $^4$He diffusion through a crystal lattice. Because the Sm contribution to the He budget is limited, this method is referred to as (U-Th)/He, with (U-Th-Sm)/He used more recently for phases with measurable Sm content (Ault et al., 2019).

Specifically, for apatite and zircon, He is well retained within the crystal structure with temperature sensitivity of ~40-120°C

and ~20-200 °C, respectively (see recent review of Ault et al. (2019) for details about the complexity of He retention). Upon

alpha decay, the alpha particle is emitted with an important kinetic energy, thus travelling along 5 to 30 µm before stopping or

being ejected from the crystal (Farley et al., 1996; Ziegler et al., 2008). Apatite and zircon (U-Th)/He ages thus require an

alpha-ejection correction for disintegrations that happen close to crystal margins (Farley et al., 1996). Apatite and zircon (U-

Th)/He dating methods then possess a specificity in demanding careful extraction of crystals from plutonic, volcanic or

sedimentary rocks using crushing, mineral separation, and crystal selection before He, U, Th and Sm analyses. Only suitable

apatite and zircon crystals should be selected for (U-Th)/He dating, then crystal size and geometry should be measured and

recorded, in order to correct the alpha loss by ejection.

In this technical note, we are focusing on the protocols developed over the more than ten years at the GEOPS laboratory,

Paris Saclay University, and more specifically on He, U, Th and Sm content determinations, while some details of the scheme

used for apatite and zircon preparation and selection are also given. Contributions that have investigated the impact on (U-

Th)/He dating of crystal quality selection, and size and shape of selected crystals, are an important source for those particular

points (e.g. Farley, 2002; Brown et al., 2013; Gautheron et al., 2012; Reiners and Farley; 2001; Reiners et al., 2005).

**2.2 Apatite and zircon samples preparation, picking and packing**

Apatite and zircon crystals are extracted from rocks by classical crushing methods, sieved (mesh $< 400\,\mu$m), and separated

following density (tribromomethane and di-iodomethane – VWR®) and magnetic methods (L-1 Frantz Isodynamic®

Separator). Mineral separation can also be performed using other, less toxic, alternatives such as lithium heteropolytungstate,

also known as LST. For the Durango apatite gem crystal, a gentle crushing in an agate mortar allows getting fragments of

different sizes. Inclusion-free automorphic apatite and zircons or Durango apatite fragments are picked under a binocular

microscope (SZX12 – Olympus®) and selected as a function of fragment size ($>60\,\mu$m). For an automorphic crystal, the

length, height, and width are measured, and the termination geometry of the crystal (broken faces, pyramids, no pyramids) is

recorded. The ejection factor ($F_T$) and equivalent sphere radius ($Rs$ or also named ESR) are determined using the Monte Carlo

simulation of Gautheron and Tassan-Got (2010); Gautheron et al. (2012) and Ketcham et al. (2011). As the ejection length for

the Th decay chain is higher than that for the U decay chain, the value of Th/U ratio is fixed to the measured value to avoid over- or under- corrections (Ketcham et al., 2011; Ziegler, 2008). An internal modification of the Monte Carlo simulation (Qt_LFT software), coupled with an Excel® automatic file generation, is used to calculate the $F_T$, $Rs$ and crystal weight from a list of different crystal geometries. The Qt_LFT software is available and added as a supplement to this technical note.

        Each apatite and zircon crystal or fragment is placed respectively into a platinum tube (99.95% purity, 1.0×1.0 mm –

Johnson Matthey®) or a niobium tube (purity 99.95%, 1.0×1.0 mm – Alfa Aesar®). The choice of the capsule metal (Nb, Pt) for packaging is strictly related to the acid attack protocol. During U, Th and Sm analysis by means of ICP-MS instruments, the presence of Pt$^+$ ions at high concentration (>320 µg/ml) in the sample solution may lead to the formation of the complex platinum argides $^{194}Pt^{40}Ar^+$, $^{195}Pt^{40}Ar^+$, $^{198}Pt^{40}Ar^+$, that cause isobaric interferences with the measured U isotopes at masses 234, 235 and 238 (Evans et al., 2005; Reiners and Nicolescu, 2007). Apatite chemical digestion can be achieved using acid digestion

at low concentration (HNO$_3$ 5N ultra-pure) and low temperature (65°C), which is not able to dissolve the capsule, therefore Pt tubes can be used. However, an acid digestion using concentrated acids (HCl and pure HF 27N) at high temperature (220°C) is instead used for zircon, which leads to a total dissolution of the Pt capsule. Therefore, Nb tubes were adopted for zircon. Although niobium-argon complexes do not create isobaric interferences with the analyzed U masses, the solution, highly concentrated in Nb, may, however, cause a partial precipitation of uranium and thorium (Evans et al., 2005). The Nb impact

on the U and Th content determination will be discussed in detail in section 3.3.

**2.3 Helium analysis protocol**

        The helium content analyses were performed at the GEOPS laboratory, Paris Saclay University (Orsay, France). Each capsule containing a crystal, fragments, or grain(s) was degassed using either a built-in-house He extraction line coupled with a quadrupole mass spectrometer (Prisma QMG 100 Pfeiffer®), further referred to as the Quad line, or another built-in-house

line connected to a rehabilitated VG5400® magnetic sector mass spectrometer, further referred to as the VG line. The Quad and VG lines are fully automated using the LabView® software, from the heating phase to the helium analysis. Each portion of the line is divided into sections (extraction, purification, analysis) by pneumatic Swagelok® valves coupled to electro-valves (E.V 3/2 NF Direct Flasque.D2,4 ALU BUNA, TH France®) and activated by pressurized air. Ultra-high vacuum conditions (pressure < 10$^{-9}$ mbar) are guaranteed by using a combination of turbomolecular (HighCube – Pfeiffer®) and ionic pumps

(StarCell – Varian®). Figure 1 presents the schematic geometry of the two built-in-house Quad and VG lines, with their different parts that are controlled using LabView®.

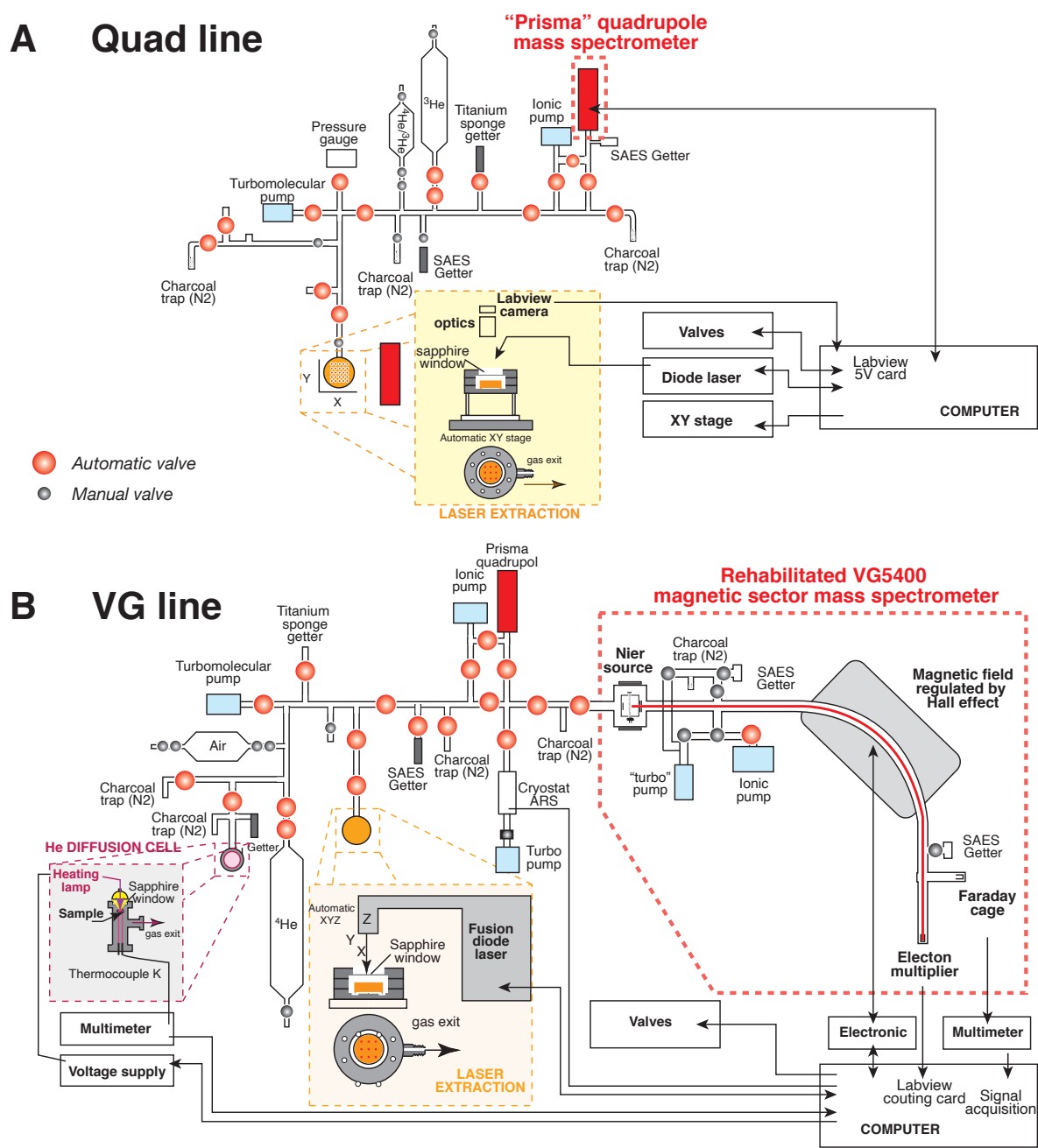

***Figure 1:*** *Schematic representation of the built-in-house systems for He extraction, purification, and analysis by mass spectrometer: (A) Quad line and (B) VG line. The part with the He diffusion cell in B is not used in this study.*

Platinum and niobium tubes adopted for sample packaging are suitable because of the low level of hydrogen they release under high vacuum, their malleability and their U, Th, Sm and REE purity requiring no pre-etching. In addition, being metallic materials, they ensure a homogeneous heat transfer during laser shooting. For the Quad line, the Pt/Nb tubes are deposited on a copper planchette having 25 or 49 positions, whereas, for the VG line, the tubes are placed on a stainless steel (inox) or a

 copper planchette with 12 or 49 positions. Copper or inox were selected for planchette material due to their good thermal conductivity and their inertia in vacuum conditions. After each sample loading, the line (Quad or VG) is heated overnight at low temperature (<50°C) using heating tapes, to remove any gas adsorbed on their inner walls.

For the Quad line, the planchette is placed into a cell that moves under the laser beam using a X-Y motor system (SMC100CC – Newport®) controlled by LabView®. An ytterbium-doped infrared (IR) diode laser (wavelength 1064 nm –

 1080 nm, power 10W, Manlight® - Laser2000®), coupled with an optic system placed at a focal distance of 4 cm from the sample, allows heating up the capsules with a beam of ~70 $\mu$m diameter. For the VG line, the heating of the tubes is ensured by an infrared Fusions Diode Teledyne® laser moving above the cell, and ultra-high vacuum can be quickly obtained by heating with the IR laser an empty capsule purposely placed on the copper/inox planchette to remove air adsorbed on it. In both cases, the cell is sealed with a CF63 sapphire window (Caburn®) allowing good transmission of the whole IR laser beam,

 but low He exchange. Each capsule is heated using the heating protocol summarized in Table 1.

**Table 1:** samples packing, and He purification and analysis protocols

| Minerals | Packing | Degassing | Purification | Mass spectrometer | Standard |
|---|---|---|---|---|---|
| **Apatite** | Pt tube | 5 min at 1050°C* | Liquid $N_2$ on activated charcoal + SAES 707 10 min at room temperature | **Quadrupole:** $H_2$, $^3$He, $^4$He, mass 5, $H_2O$, $^{40}$Ar, $CO_2$ Electron multiplier 850 V | Volcanic Durango apatite 31.02±1.01 Ma; McDowell et al. (2005) |
| **Zircon** | Nb tube | 30 min at >1150°C * | Liquid $N_2$ on activated charcoal + SAES 707 10 min at room temperature | **VG5400:** $^4$He Electron multiplier 3500 V + LabView ion counting | Volcanic Fish Canyon Tuff zircon 28.5±0.06 Ma; Schmitz and Bowring (2001) |

* apatite and zircon can be heated at different temperatures and with different time lengths, especially zircon that should be heated at higher temperature. Equations established by Fechtig and Kalbitzer (1966) can be used to calculate the minimal time, for a given temperature, that should be used to ensure complete He degassing, knowing the He diffusion coefficient. It is, however, important to not heat apatite crystals at too high temperature to reduce any issue with U or Th volatilization.

The heating schedule procedure is repeated on each sample until all $^4$He is degassed, giving a signal back to the background level within less than 2%. The sample temperature achieved using the laser of the Quad line is recorded by means of a LabView® camera and a built-in-house algorithm that converts the total red, green, and blue visible light into a temperature. We use the visible light emission part of the black body light emission during heating. To this aim, a Pt capsule was heated with increasing values of laser intensity and pictures were taken at different temperatures, as presented in Figure 2A. At the same time, the temperature of the heated capsule was measured using an external filament extinction pyrometer. This type of pyrometer is currently used to calibrate TIMS filament temperatures. For each recorded picture, a simple image treatment has been realized using LabView® to retrieve the red, blue, and green value on a RGB colorimetric coding system that ranges from 0 to 255 (Fig. 2B). As the red signal is already saturated when the capsule is emitting visible light, we chose to sum the signals of the three colors, as shown on Figure 2C. The obtained RBG sum is correlated with the temperature of the heated capsule. This simple image treatment procedure has been calibrated from 950 to 1150°C for a fixed value of the exposure time of 500 ms. The same image treatment has been automatically applied to each heated capsule, allowing to retrieve temperature, with an estimated error of ± 20°C on temperature calculation.

The protocol has been built for the purpose of degassing apatite crystals or fragments that are packed in a Pt tube, where the chosen heating time and temperature schedule should permit to retrieve all the He from apatite during the first degassing step, as predicted using the mean diffusion coefficient of Farley (2000) for Durango apatite. If the unknown apatite is pure and presents a diffusion coefficient similar to the Durango apatite, the He content of the second step should be similar to background. This protocol thus allows to detect any He retentive mineral inclusion (e.g., titanite, zircon) that was not seen on picking and contributes to the He budget (Farley, 2002). It can also be used to monitor the impact of radiation damage on the retentiveness of an apatite crystal. For zircon, as mineral inclusions are not an issue for (U-Th)/He age interpretation, we simply heat the Nb capsule at a temperature close to image color saturation (~1200°C; Fig. 2) and ensure to avoid any

overheating problem such as capsule melting. Because Pt and Nb have different emissivity values, the temperature calibration is not totally adapted to Nb. Indeed, the color visible for a given temperature is not the same for Nb and Pt capsules, and the real temperature of a Nb capsule should slightly differ that displayed. Nevertheless, the zircon crystals can be heated until total He degassing, and neither the poor determination of heating temperature nor the different emissivity values are an issue.

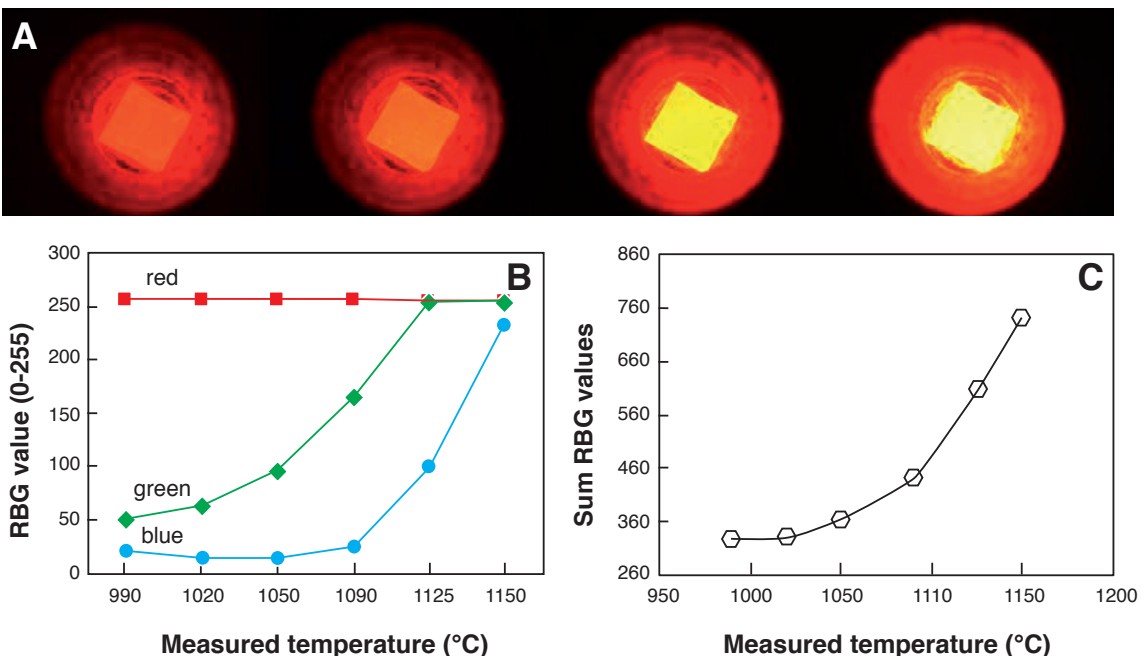

***Figure 2:*** *temperature calibration procedure of a heated Pt capsule in visible light.* ***A)*** *Example of capsules heated with different laser beam intensities showing the change in color.* ***B)*** *and* ***C)*** *Evolution, as a function of the temperature measured using a pyrometer, of the RBG code values and their sum, respectively.*

The analysis protocols differ as a function of the type of mass spectrometer used and the type of analyzed minerals.

### 2.3.1 Quad line

The diffused ⁴He gas is mixed in the purification line with a known amount of ³He, used as a spike, in concentration of about 100 to 1000 times higher than the ⁴He to be determined. A ~4000 cc (cubic centimeter) cylinder (V1), filled by ³He gas, is connected to a pipette made by two welded valves with a small, 5 mm diameter, stainless steel cylinder placed inside to reduce the volume of the pipette (Fig. 1). The approximate volume of the pipette is ~0.5 cc (V2) and allows an ³He amount of

$10^{-9}$ to $10^{-10}$ ccSTP (cubic-centimeters-at-standard-temperature-and-pressure) to be introduced into the line. The amount of $^3$He decreases in the cylinder by a factor of ~0.9999 (V1/(V1+V2)) for every shot of gas extracted, therefore the number of pipettes taken is automatically recorded to take the decrease of $^3$He in the cylinder into account. These statistics allow the data for He age computation to be calculated with the right amount of $^3$He spike introduced into the line (see also below). According to the spike conditions reported above ($^3$He in concentration of about 100 to 1000 times higher than $^4$He), it has been observed that the Quad line can perform a total of about 6000 analyses. After which, the $^3$He gas tank is refilled using a 1 L tank filled with $^3$He (99.5% - Eurisotop®) at 1 bar.

The sample gas is purified from most of the $H_2O$, $CO_2$, $H_2$, Ar gases using two liquid nitrogen-cooled traps of activated charcoal, a St707 SAES® getter operated at room temperature, according to different purification protocols adapted for various minerals (Table 1). The use of a hot (>850°C) titanium sponge getter (3-13 mm, 99.95% metal basis - Alfa Aesar®) is dedicated to minerals with high $CO_2$ or $H_2O$ contents. The access to the entire system of traps, individually connected to the line by means of ultra-high vacuum valves (Fig. 1), allows the analysis of a large variety of minerals, containing variable abundances of $CO_2$ or $H_2O$, such as calcite or goethite (Allard et al., 2018; Cros et al., 2014). Beside helium isotopes ($^3$He and $^4$He), $H_2O$, $CO_2$, $H_2$ and Ar gases are additionally measured on the electron multiplier of the Prisma QMG 100 Pfeiffer® quadrupole mass spectrometer to check for the effective purification of the analyzed gas. Such measurements of the gas are repeated 16 times and a linear regression of the data for the $^4$He/$^3$He ratio is then calculated and includes a correction of the HD$^+$ isobaric contribution to the $^3$He signal, even if this contribution is insignificant compared to the $^3$He spike signal. In addition, we also observe that the signal at mass 4 slightly increases when the $H_2$ signal is higher, which we interpret as either a double $H_2$ molecule having an isobaric impact on mass 4 or the tail of the $H_2$ peak having an influence on the shape of the mass 4 peak. This effect is not negligible for the low $^4$He signals of typical samples, but an adapted $H_2$ purification protocol allows to remove this effect: a St701 SAES® getter unit is positioned just ahead of the quadrupole mass spectrometer (Fig. 1). Longer gettering time could be offered as a suitable alternative to reduce the $H_2$ influence on the $^4$He peak, if addition of a getter to the mass spectrometer is impossible.

The gas purification protocol, combined to $^3$He spiking, ensures to get a close-to-constant total pressure in the line and in the quadrupole mass spectrometer. The $^4$He abundance is calculated from the introduced $^3$He amount such as:

$$^4He = \left(\left(\tfrac{^4He}{^3He}\right)_s - \left(\tfrac{^4He}{^3He}\right)_b\right) \times\ ^3He \tag{1}$$

with $(^4He/^3He)_s$ and $(^4He/^3He)_b$ the isotopic ratios measured for the sample and blank respectively. The $^3He$ amount is determined from the calibrated value, noted $^3He_c$, the two volume values (V1 and V2) and two evolution parameters (N and D), using the following equation:

$$^3He =\ ^3He_c \times \left(\tfrac{V1}{V1+V2} \times D\right)^N \tag{2}$$

where V1 is the volume of the $^3He$ cylinder (~4000 cc), V2 the pipette volume (~0.5 cc), N the pipette number (i.e., N is the number of introductions of the pipette volume), and D is the 'drift', an additional parameter introduced to account for the evolution of the sensitivity of the quadrupole mass spectrometer along with external parameters such as temperature or even power failures. Both parameters $^3He_c$ and D are obtained empirically by calibration from several analyses of Durango apatite fragments. D acts as if the pipette volume V2 could vary to mime the variations of the quadrupole sensitivity and is updated manually according to the results of the Durango standard, which is analyzed regularly, especially every time the source is tuned again. The product $^3He_c \times \left(\tfrac{V1}{V1+V2} \times D\right)^N$ thus decreases regularly and homogeneously.

**2.3.2 VG line**

The diffused $^4He$ is purified from the $H_2O$, $CO_2$, $H_2$, Ar gases using two St707 SAES® getters and a Ti sponge getter (Fig. 1B). A cryogenic trap from Advanced Research Systems® (ARS), installed more recently, has the capacity to cool activated charcoal down to 8 K. At this temperature, He is efficiently trapped, then further released into a smaller volume at about 50 K. Again, according to the nature of the sample to be analyzed, different purification protocols are adopted (Table 1). Every protocol is fully automatized and the $^4He$ gas is introduced into the VG5400® magnetic sector mass spectrometer. The filament amperage is chosen to obtain a compromising value for the trap current of 300 to 400 μamps, lower than the recommended value for He analysis but ensuring a longer life to the filament. Isotope $^4He$ is analyzed by ion counting using a Pfeiffer® electron multiplier (17 dynodes) connected to an Ortec® discriminator and a LabView® counting card. 20 analyses of the $^4He$ signal integrated over 1 sec are performed and the mean $^4He$ signal is recorded (while a linear regression is available, only the mean signal and associated standard deviation are recorded). The dead time of the electronic chain is close to the width of the pulse delivered by the electron multiplier, which is a few ns. The maximal recorded counting rate being about

$3 \times 10^5$ cps (counts per second), the dead time correction is always lower than 1% and it is neglected. The system sensitivity is empirically determined from [4]He analyses of the Durango standard (see section 3.2), while the internal [4]He standard from a ~4000 cc cylinder is only used to check for sensitivity change over time and signal reproducibility. The [4]He abundance in a sample is computed using

$$4_{He} = \left(4_{He_s} - 4_{He_b}\right) \times S \tag{3}$$

where [4]He$_s$ and [4]He$_b$ are the signals measured for the sample and blank respectively, and $S$ is sensitivity in ccSTP He/cps; cps: counts per second. The unit ccSTP is not the SI unit for amount of substance, but a way of expressing gas moles by the volume (cubic centimeter) they would occupy at the standard temperature of 273,15 K = 0°C and the standard pressure of 101,325 Pa = 1 atm, which does not follow the 1 bar pressure 1982 IUPAC recommendation, but is historically used by the noble gas community and most (U-Th)/He geochronological users. We thus keep this definition of standard pressure and temperature conditions, STP, where 1 mole occupies 22,414 ccSTP.

Durango apatite fragments and/or Fish Canyon Tuff zircon crystals are analyzed regularly (1 Durango/Fish Canyon standard analyzed every 7 unknown samples) to check the (U-Th)/He analysis reproducibility.

## 2.4 Digestion chemistry protocol

### 2.4.1 Vessel cleaning

For apatite, we used single-use 4 ml polypropylene (PP) snap-cap tubes (supplier VWR®) that do not need prewashing for our purpose (Table 2).

For zircon, the dissolution is made in 350 $\mu$l PFA (PerFluoroalkoxy Teflon®) parrish style vials (Savillex®) placed into a high pressure-high temperature dual wall digestion vessel. Before their use for acid digestion of samples, the vials undergo a series of acid baths in a 250 ml beaker (borosilicate glass – VWR®) placed on a hot plate at 100°C, according to the following sequence: cycle 1: 24h bath in diluted 5% Extran® MA 02 (Merk®) in Milli-Q® water (Milli-Q® HX 7000 SD); cycle 2: 24h bath in HCl 5N (Emsure® 32% VWR®); cycle 3: 24h bath in HCl/HNO$_3$ 3/1 (HCl Emsure® 32% VWR®; HNO$_3$ Emsure® for analysis 65% VWR®); cycle 4: 24h bath in HNO$_3$ 5N (Emsure® 65% VWR®). Between each bath, vials are rinsed with Milli-Q® water (18 MΩ Direct 8 System – Merck® Millipore®). Vials are finally dried in an oven at 50°C and stored in a hermetically closed PP box until further use (Table 2).

To check their degree of cleanness after the series of cleaning baths, cleaned vials are filled with 200 $\mu$l Milli-Q® water, refluxed for 2h at 100°C, and this water is transferred to PP tubes with the addition of 800 $\mu$l MilliQ® water, for analysis by ICP-MS of $^{238}$U and $^{232}$Th natural isotopes and spike isotopes $^{230}$Th and $^{235}$U.

### 2.4.2 Spike solution composition and calibration

According to the isotopic dilution method, for every apatite sample to be analyzed, a volume of 50 $\mu$l of spike, the spike solution MR2, is introduced into the vial before the dissolution protocol (100 $\mu$L for zircon). This spike solution is prepared (60 ml) every 6 to 12 months from elemental mother solutions MR1 and MR (Appendix A, Table), these being obtained from certified concentrated mono-elemental solutions: for MR, we mix $^{235}$U (10 ml, IRRM-50, 4.2543(11) nmol $^{235}$U.g$^{-1}$, EU JRC - Belgium), $^{230}$Th (5 ml, IRRM-61, 2.474(18) nmol $^{230}$Th·g$^{-1}$, EU JRC - Belgium) and $^{149}$Sm (100 ml, 7 ppm, 90%, 47,01 nmol $^{149}$Sm.g$^{-1}$, Berry and Assoc® - MI, USA). For apatite analysis, $^{42}$Ca (2.44 mg solid $^{42}$CaCO$_3$, 97.8%, BuyIsotope® - Sweden) is added. Detention of EU JRC materials is authorized under Laboratory License for Radioactive Materials (RMs). Other isotopes, such as $^{233}$U or $^{229}$Th, are also available as isotopic spikes at JRC; however, their high cost due to the radioactive nuclear materials transport fees is not negligible; for this reason, we chose to use isotopes $^{235}$U and $^{230}$Th. In over more than ten years of research at the GEOPS laboratory, 16 solutions of spike for apatite and 3 solutions of spike for zircon have been produced. Since the early development of the (U-Th)/He dating method at GEOPS, other isotopes than $^{235}$U and $^{230}$Th were recently added to the protocol, namely $^{149}$Sm and $^{42}$Ca. For zircon, the addition of the isotope $^{149}$Sm to the spike solution was discarded due to the negligible impact of $^{147}$Sm decay on the He budget compared to the He production from $^{238}$U and $^{232}$Th.

Before use, the spike solution MR2 is calibrated for U, Th, Sm and Ca, using a series of weighted solutions, each of them obtained by mixing a volume of another mono-elemental standard solution with a volume of the spike MR2 (Appendix A, Table): the U$_{SS}$ solution contains an $^{238}$U mono-elemental solution at 1015 ppm (99.96%, Analab® France), diluted and mixed with the $^{235}$U spike; the Th$_{SS}$ solution contains a $^{232}$Th mono-elemental solution at 993 ppm (99.93%, Analab® France), diluted and mixed with the $^{230}$Th spike; the Sm$_{SS}$ solution contains a natural Sm solution at 1006 ppm ($^{147}$Sm = 14.99%, $^{149}$Sm = 13.82%, Analab® France), diluted and mixed with the $^{149}$Sm spike; and the Ca$_{SS}$ solution contains a natural Ca solution at 1003 ppm ($^{42}$Ca=0.65%, $^{43}$Ca=0.13%, Analab® France) mixed with the $^{42}$Ca spike. These solutions are then analyzed by ICP-MS to properly calibrate the MR2 spike solution.

### 2.4.3 Samples digestion protocol

After degassing, each Pt/Nb-conditioned sample is transferred from the planchette into a vial for grain dissolution by acid digestion. Sample digestion is started by adding in each vial a volume of 50 $\mu$l for apatite (or 100 $\mu$l for zircon) of a spike solution (in $HNO_3$ 5N and containing a known amount of $^{235}U$, $^{230}Th$ - plus addition of $^{149}Sm$ and $^{42}Ca$ for apatite). According to the nature of the sample, a specific dissolution protocol is followed (acids and heating temperature) (Table 2).

**Table 2:** Cleaning chemistry and chemistry protocols for prepared solution

| Mineral | Vial type | Cleaning protocol | Spikes | Dissolution protocol | Solutions |
|---|---|---|---|---|---|
| Apatite | 4 ml single use PP tube | No cleaning needed | 50 $\mu$l ($^{235}U$ ~4 ppb; $^{230}Th$ ~4 ppb, $^{149}Sm$ ~4 ppb; $^{42}Ca$ ~1 ppm) | + 50 $\mu$L 5N $HNO_3$ and heating for 3 h at 65°C. + cooling time (30 min) + 1.9 ml 1N $HNO_3$ | **Blk:** acid blank<br><br>**Blk-ch:** acid blank chemistry<br><br>**BSP:** spiked blank |
| Zircon | 350 $\mu$l PFA tubes + PFA container + stainless steel digestion bomb + oven + 4 ml single use PP tube | PFA vials on a hot plate at 100°C: Extran 5% bath (24h); HCl 5N Emsure bath (24h); HCl:$HNO_3$ (1:3) Emsure bath (24h); $HNO_3$ 5N Emsure bath (24h) | 100 $\mu$l ($^{235}U$ ~45-55 ppb; $^{230}Th$ ~15-20 ppb) | **Step 1:** Vial: +200 $\mu$L HF 27N + few drops $HNO_3$ 7N vessel: 10 ml HF 27N + 1 ml $HNO_3$ 7N oven: 220 °C for 96h **Step 2:** Complete evaporation at 100 °C **Step 3:** Vial: + 300 $\mu$L HCl 6N; Vessel: 12 ml HCl 6N; Oven: 220°C for 24h **Step 4:** Complete evaporation at 100°C **Step 5:** Add 200 $\mu$L $HNO_3$ 5N + some drops HF 0.1N; 1h reflux at 100°C | **BSP-ch:** spiked blank chemistry<br><br>**BSP-ch-Pt:** spiked blank chemistry with a Pt capsule for apatite<br><br>**BSP-ch-Nb:** spiked blank chemistry with a Nb capsule for zircon<br><br>**DUR:** natural Durango<br><br>**Sp:** spiked sample |

| | | | | **Step 6:** Transfer to PP vial, add 800 $\mu$L HNO$_3$ 1N  **Step 7:** Transfer to PP vial, dilution 1/10 with HNO$_3$ 1N | including standard |
|---|---|---|---|---|---|

275     For apatite, the dissolution protocol (Table 2) was adapted from Farley (2002). Dissolution requires a soft acid digestion (HNO$_3$ 5N – bidistilled from HNO$_3$ 65– Normapur® – VWR®) performed in a 4 mL single-use polypropylene tube (VWR®) by adding 50 $\mu$L of spike (~4 ppb of $^{235}$U, $^{230}$Th, $^{149}$Sm and ~1 ppm $^{42}$Ca) and 50 $\mu$l of HNO$_3$ 5N. The tube is then placed on a hot plate at 65°C during 3 h for digestion. After digestion and cooling, samples are diluted with 1.9 ml of HNO$_3$ 1N and stored at 4°C before ICP-MS analysis (Table 2). Due to the digestion conditions (using diluted acids), the Pt capsule does not dissolve

280 and does not interfere with ICP-MS analysis. Sample digestion is always made with nitric acid, freshly diluted by addition of Milli-Q® water. To minimize possible contamination sources from the environment, storing tubes, or evaporation of the solutions, ICP-MS analysis is scheduled within a few days of sample digestion. After analysis, the Pt capsules are promptly collected, cleaned, and sent back to the factory company in a recycling/reselling loop contract.

    For zircon, the dissolution protocol was slightly adapted from Reiners (2005) and Reiners and Nicolescu (2007). The

285 dissolution is performed in 350 $\mu$l PFA parrish style vials (Savillex®). Zircons are first spiked with 100 $\mu$L of $^{235}$U and $^{230}$Th (~45 to ~55 ppb of $^{235}$U, ~15 to ~20 ppb of $^{230}$Th; Table 2). The vials are then placed into a digestion vessel hermetically sealed with a metallic gasket to hold high pressures (IN/PFA-OUT/Stainless steel PA4748, Parr Instrument Company®). The digestion follows several steps summarized in Table 2. Step 1: inside the vials: addition of 200 $\mu$L HF 27N (Suprapur® – VWR®) and few drops of HNO$_3$ 7N (Suprapur® – VWR®); inside the digestion vessel: addition of 10 ml HF 27N and 1 mL of HNO$_3$ 7N. Once sealed, the vessel is heated up at 220°C in an oven and held at high pressure for 96 hours. Step 2: the acid

290 solution is evaporated to dryness by placing the vials on a hot plate at 100°C. Step 3: 300 $\mu$L HCl 6N (Suprapur® – VWR®) are added to each vial, the Parr vessel is filled with 12 ml HCl 6N, sealed and heated back at 220°C, under pressure for 24 hours in an oven. Step 4: the vials are evaporated to dryness on a hot plate at 100°C. Step 5: a reflux is carried out with a combination of HNO$_3$ 5N (200 $\mu$L) and HF 0.1 N (few drops) at 100°C for 1 hour. Step 6: the solutions are transferred to 4 ml

295 polypropylene tubes where 800 $\mu$L HNO$_3$ 1N are added. Step 7: a final dilution (1/10) is done with freshly prepared HNO$_3$ 1N

in a second 4 mL PP single-use tube. The solutions are then stored at 4°C before isotopic analysis. To avoid pollution released from the storage tubes or changes in concentration by evaporation of the solutions, the ICP-MS session is always scheduled within a few days after samples dissolution.

In addition to the apatite and zircon samples, the following types of solution are also prepared, as summarized in Table 2: spiked sample (Sp), including Durango and FCT standards; acid blank (Blk): to check acid purity and potential contaminations of tubes; chemistry acid blank (Blk-ch): to check the enrichment contamination in acid caused by the chemistry protocol; spiked blank (BSP): a weighted volume of spike is added to a volume of acid in order to check the variations in concentration of the spike and to take the contribution of natural isotopes contained in the spike into account; spiked blank chemistry (BSP-ch): a volume of spike in acid undergoes the same dissolution protocol than the samples, which allows to quantify contamination coming from contingencies during the dissolution protocol (vessel, user, acid); Durango solution (DUR): a single fragment of Durango is dissolved in a volume of acid, no spike is added, which allows the natural isotopic ratio of uranium to be measured for checking, as it has to be in isotopic equilibrium, with $^{238}U/^{235}U=137.88$ (or $^{235}U/^{238}U=0.00725268$).

## 2.5 U, Th (Sm and Ca) analysis by ICP-MS

The solutions obtained by chemical dissolution of the samples are then analyzed with an ICP-MS to determine the U, Th and Sm (Ca for apatite) signal intensities. Since 2016, we mainly use a High Resolution Inductively Coupled Plasma Mass Spectrometer (HR-ICP-MS - ELEMENT XR® from Thermo Scientific®) at the GEOPS laboratory, that allows the U, Th and Sm isotopes to be measured at low resolution (300) while Ca is measured at high resolution (10,000). In addition, we also use a quadrupole ICP-QMS seriesII CCT Thermo-Electron® at LSCE (Gif/Yvette; France) and an Agilent® 7900 quadrupole ICP-MS at IPGP (France) to measure the U, Th and Sm contents.

The U, Th, Sm and Ca abundances are then deduced from the measured $^{235}U/^{238}U$, $^{230}Th/^{232}Th$, $^{149}Sm/^{147}Sm$ and $^{42}Ca/^{43}Ca$ isotopic ratios and equations derived from Evans et al. (2005), by means of a built-in-house Excel® workbook that is available upon request to A.D., based on the VBA automation software. Appendix B presents the equations used for the U, Th, Sm and Ca content determination. The measurement of the Ca content and the deduced apatite crystal weight, combined to the measurements of U, Th, and Sm abundances, then allow the U, Th, and Sm concentrations of the Durango fragments to be

determined. Determination of crystal weight is also useful to ensure that the criteria imposed for grain selection, i.e. (i) crystal

size (L, W, T > 60 $\mu$m), (ii) geometry (well-shaped prisms), and (iii) purity (inclusion-free crystals) have been respected.

**2.6 (U-Th)/He age reduction**

For a sample, the (U-Th)/He age (in Ma) is calculated assuming a linear production of [4]He with time, using the determined

U, Th and Sm abundances and the equations presented in Appendix C. The assumption that the [4]He production is linear holds

for ages lower than ~150 Ma, since the half-lifes of the U, Th and Sm isotopes are ~5 to ~600 times higher than 150 Ma. When

the age is older than 150 Ma, we use a trial-and-error code to calculate the age, that turns out to change by never more than a

few percent.

For an ICP-MS session, the (U-Th)/He data reduction Excel® workbook can calculate the U-Th-Sm concentrations (in

ppm by mass) and their associated standard deviations for the different minerals analyzed, according to the chemical

dissolution protocol followed, as well as the effective uranium content, eU (in ppm), the Th/U and Sm/Th ratios, and the (U-

Th)/He ages.

**3 (U-Th)/He results and discussion**

**3.1 Helium quadrupole analysis**

One aspect of quadrupole mass spectrometry is the variable response in terms of ionization and signal generation of such

instruments. This behavior has also been observed on the quadrupole adopted here for analyses of rare gases and is illustrated

on Figure 3A. The signal of the [3]He spike is reported as a function of the number of extracted pipettes, over a period of 5

months of analysis. The [3]He signal fluctuates significantly although the [3]He amount in each pipette should decrease smoothly

following a law that depends on the volume of the cylinder and the pipette volume (here ~4000 and ~0.5 cc, respectively; see

equation (2)). The advantage of using a [3]He spike for isotopic dilution is to thwart the impact of the nonlinear answer of the

quadrupole mass spectrometer with time (Farley, 2002; House et al., 2000), but it also allows the total gas pressure to be

buffered in the mass spectrometer if the introduced [3]He signal is large enough compared to the other signals.

Figure 3B presents the various signals measured with the quadrupole mass spectrometer ($H_2$, [3]He, [4]He, [40]Ar, $CO_2$, and

mass 5 that represents the background noise) during the same 5 months of analyses (~10 to 30 analyses per day, 5 days a

week). The $^3$He clearly controls the total pressure inside the mass spectrometer during the analysis, independently from the
pressure of $^4$He gas released from the samples. The use of a spike rich in $^3$He allows to get a stable and uniform total pressure in the mass spectrometer, for any degassed sample analyzed. In addition, we are following the $H_2$, $CO_2$ and Ar signals to guard against any leak and gas purification problem. For example, on figure 3B, the $H_2$ signal was higher than usual at pipette number of 200 to 250, alerting about a technical problem related to purification. It will then help to take the AHe ages with caution for low $^4$He samples, considering the $H_2$ tail can influence the $^4$He signal (section 2.3.1 above).

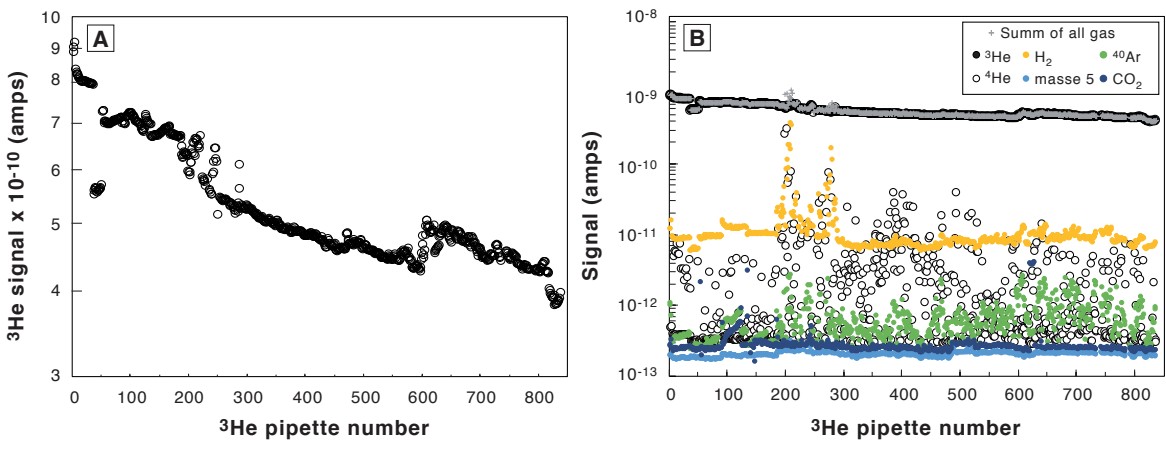


***Figure 3:*** *Evolution of the measured $^3$He signal with pipette number (from January to June 2019). (A) Decrease of the spike $^3$He signal. (B) Evolution of all the measured signals ($^3$He: black dots; $^4$He: open black dots; $H_2$: orange dots; mass 5 (background noise): light blue dots; $^{40}$Ar: green dots, and $CO_2$: dark blue dots; crosses: summation of all the signals).*

However, the quadrupole signals are variable and sometimes erratic within a few percent of the signal over weeks to months periods of time (Fig. 3A and 4A). To correct for this quadrupole drift, we introduce a correcting factor, noted D, and a calibrated $^3$He value, noted $^3$He$_c$, in the calculation of the $^3$He pipette content (equation 2), that we both adjust using the deviation of the Durango ages obtained over a period of 1-2 months. Parameter D can be viewed as modifying the value of the pipette volume, V2 in equation (2), to counterbalance the variations of the quadrupole sensitivity. As we determine the U, Th
(Sm and Ca for apatite) contents every 1 to 2 months, we also determine the D and $^3$He$_c$ values for each batch of analyses (unknown samples and Durango standards) when we are calculating the (U-Th)/He ages.

Figures 4C and D presents the Durango apatite (U-Th)/He ages obtained from measurements over a period of two months using two coupled values of the drift parameter, D, and the $^3He_c$ calibrated content. We choose to present this example, to explain how we are doing the calibration and because of the presence of age outliers. Firstly, one can observe that the (U-

Th)/He ages obtained for the Durango apatite remain constant and are more reproducible using a D value of $99.937 \times 10^{-2}$ as compared to $99.987 \times 10^{-2}$. In the present case, it just depletes the tank in $^3He$ more quickly. We are considering, over the analysis batch time (here pipette number 300 to 525), the variation of the $^3He$ signal and the lowest standard deviation of Durango AHe ages to determine empirically the D and $^3He_c$ values to be used (Fig.4A, B, C and D). Each couple of values of the $^3He_c$ and D parameters are recorded, allowing to detect any problem during the measurements. Most of the time, the

sensitivity is not evolving and the same D and $^3He_c$ values are used. However, each time the filament or the voltage of the multiplier are changed, those values need to be tuned. This protocol allows to get a better reproducibility for ages of Durango and unknown samples by 1 to 3 %. This is not significant in comparison to the dispersion of AHe and ZHe ages of most natural samples but allows to get more reproducible ages. In addition, this way of determining the mass spectrometer sensitivity makes precise knowledge of the volumes of the pipette system (V1 and V2) not so important.

Secondly, we are also checking for any analytical issues the apatite (U-Th)/He ages, including Durango apatite, during the analysis batch. As observed on Fig. 4C-D, three Durango ages are lower or higher than the expected age of $31.02 \pm 1.01$ Ma from McDowell et al. (2005). Those outliers can indicate either that the Pt tube was too closed, preventing the acid from penetrating into the closed tube, or an issue during acid digestion, or an analytical issue during ICP-MS analysis, or a too high heating temperature that has vaporized U and Th, or any issue during He quadrupole analysis. Because, in quadrupole gas

analysis, we are recording the tube temperature during heating of sample and backgrounds, the issue should be related to a digestion problem or the ICP-MS analysis. For the cases presented on Fig. 4C-D, the lower ages were associated with Th/U ratios that are significantly different from other aliquots, tracing the incomplete recovery of U, Th and Sm associated with non-total digestion or clogging of the solution during ICP-MS analysis. We thus check, for unknown samples, any digestion or ICP-MS issue.

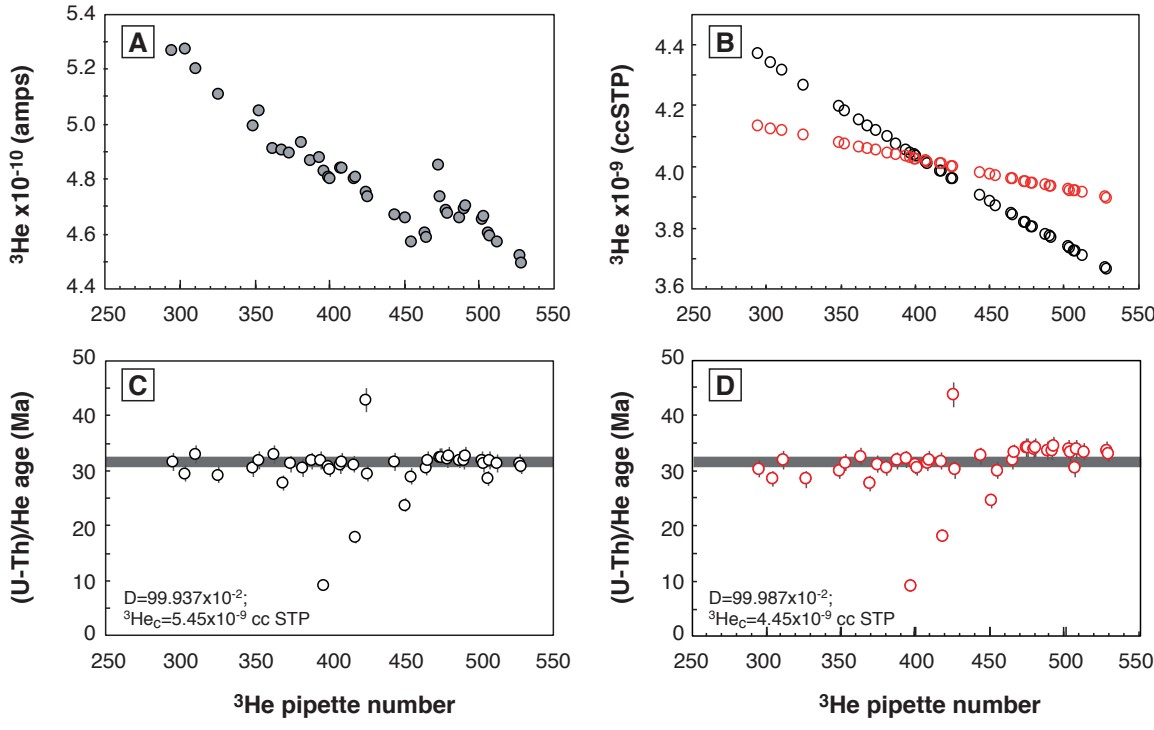


***Figure 4:*** *Evolution of the Durango apatite age as a function of pipette number.* ***(A)*** *Evolution of the ³He signal over the period of analysis.* ***(B)*** *Evolution of the calculated ³He content of each pipette for two different D drift values (black dots or red dots) and ³He$_c$ calibrated content, using equation (2).* ***(C)*** *and* ***(D)*** *calculated Durango AHe ages for the different D drift values and ³He$_c$ calibrated content.*


In addition to the ³He pipette number associated to each measurement, we use a specific code name. As an example, for a Durango apatite, the code name D19P11A can be read as Durango, year 2019, planchette n°11, aliquot A. This designation using the name of the sample and its analysis year allows to better organize the He analyses database and data backup from a chronological point of view.

**3.2 Helium magnetic sector analysis**

In comparison to quadrupole mass spectrometers, magnetic sector mass spectrometers have a more stable and linear response of ionization and thus allow for better analysis. To test the response of the modified VG5400® mass spectrometer, determine the ⁴He sensitivity and calibrate the ⁴He cylinder, we perform multiple analyses of fragments of Durango apatite

having different sizes (dozens to hundreds of micrometers long fragments). After degassing, the U, Th and Sm contents (and

AHe ages) of the fragments are determined and, assuming an age of 31.02±1.01 (McDowell et al., 2005), the amount of $^4$He

is calculated for each fragment. The calculated $^4$He contents are compared to the measured $^4$He signals in Fig. 5A, showing a

correlation between the calculated $^4$He in ccSTP and the $^4$He signal in count per second (cps), allowing the sensitivity to be

determined. The ratio of the calculated $^4$He to the measured $^4$He normalized to the obtained sensitivity is plotted versus

measured $^4$He in Fig. 5B. This operation was performed in 2016 and 2018, as the VG5400® was tuned with different source

parameters and using different line conditions: since 2018, we are using a cryogenic trap allowing to concentrate the gas in a

smaller volume and then increase sensitivity (Fig. 5A).

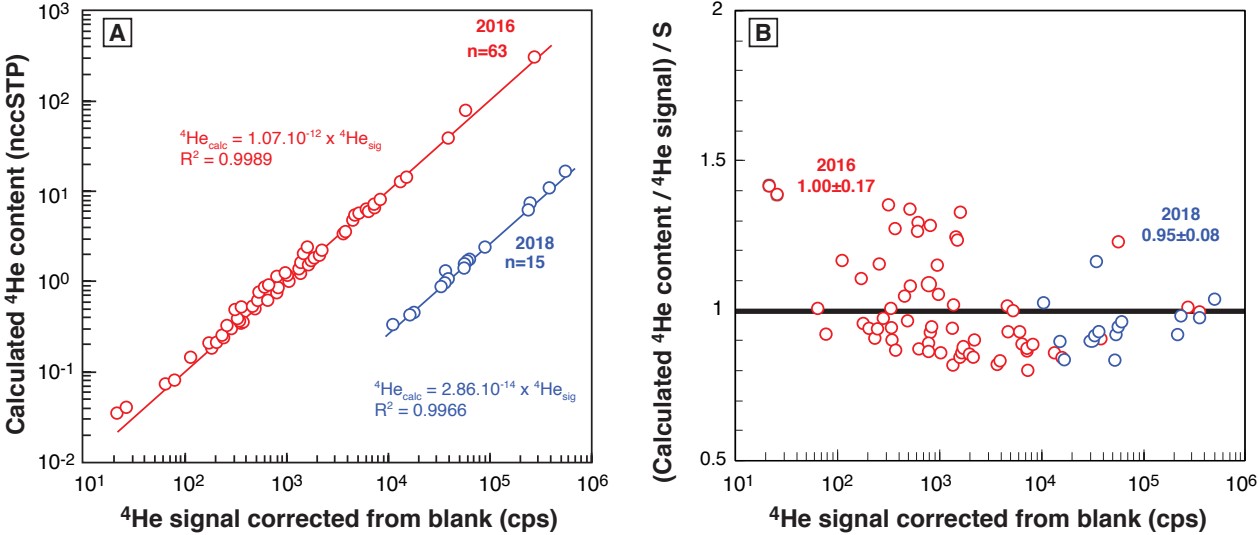

*Figure 5: VG5400® magnetic sector mass spectrometer sensitivity determined using fragments of Durango apatite. (A) log-log diagram presenting the calculated $^4$He content (nccSTP) versus the measured $^4$He signal (cps), for two determinations in*

*2016 and 2018. (B) Ratio of the calculated to the measured $^4$He normalized to the obtained sensitivity (S) versus measured $^4$He for both datasets. In 2016, a He sensitivity of $1.1 \times 10^{-12}$ ccSTP He/cps was obtained. In 2018, the addition of a cryogenic trap and modifications of source parameters allowed to get a better sensitivity of $2.9 \times 10^{-14}$ ccSTP He/cps (cps: counts per second). n is the number of analyzed fragments.*

For the two different conditions, the measured $^4$He signals and the $^4$He contents calculated from measured U-Th-Sm

display a very good linear correlation ($r^2>0.99$, Fig. 5A). Derived sensitivity values, S, are $1.1 \times 10^{-12}$ ccSTP He/cps in 2016 and

$2.9 \times 10^{-14}$ ccSTP He/cps in 2018, additionally showing that the use of the cryogenic trap increases the sensitivity by a factor ~40. Fig. 5B also reveals the dispersion of the calculated $^4$He content. For the 2016 dataset, a large range of Durango apatite fragments were selected, including smaller fragments of <50 $\mu$m in size, to get a large range of $^4$He signals. However, this may

introduce difficulty associated with U, Th and Sm content determination in very small grains. The dispersion of the normalized ratio of calculated content to He signal observed for low $^4$He signal (Fig. 5B) may reflect the difficulty to determine the U-Th content for very small Durango fragments. For the 2018 sensitivity calibration, we selected larger Durango fragments that reduce data dispersion (Fig. 5B).

Such analyses of Durango apatite fragments are also useful to test the electron multiplier and counting system responses,

which turned out to be linear from thousands to hundreds of thousands of cps for He, without any impact of the dead counting time. However, we are always trying to select standard fragments yielding signals similar to unknown samples. Further, the use of both a $^4$He cylinder and Durango apatite fragments is important to follow the evolution of the filament and analyzer conditions through time. The $^4$He tank thus gets calibrated and is mainly used to check for signal reproducibility change and analyzer stability. In addition, Durango apatite fragments are analyzed regularly to follow the sensitivity evolution though

time, particularly after each power failure, that occasion slight modifications of the filament emission condition.

### 3.3 U, Th, Sm chemistry and blanks

Acid blanks are regularly analyzed allowing the acid quality to be controlled. Low intensities are measured for $^{235}$U and $^{230}$Th, i.e. <20 cps, meanwhile the intensities for $^{238}$U and $^{232}$Th are hundred times higher. Higher signal intensities for all the isotopes are measured in Blk-ch, indicating that the chemical dissolution method adds some contamination to the sample

solutions. Such contribution is nevertheless very negligible compared to the intensities of the signals observed for the apatite and zircon samples (100,000 cps).

For apatite, U, Th and Sm in blanks are low in comparison with the U, Th, Sm contents of the apatite, as already stated by Reiners and Nicolescu (2007). Figure 6 presents the evolution of the measured $^{235}$U/$^{238}$U, $^{230}$Th/$^{232}$Th and $^{149}$Sm/$^{147}$Sm isotopic ratios for spiked blank (BSP), spiked blank chemistry (BSP-ch) and for Durango apatite sample (Sp) solutions. For a spiked

sample (Sp), the $^{235}$U/$^{238}$U, $^{230}$Th/$^{232}$Th and $^{149}$Sm/$^{147}$Sm ratios range between the BSP-ch value and the natural value (i.e. $^{235}$U/$^{238}$U=0.00725268; $^{230}$Th/$^{232}$Th=0 (no natural $^{230}$Th atoms) and $^{149}$Sm/$^{147}$Sm=0.9), as can be observed in Figure 6. The

isotopic ratio values for the BSP and BSP-ch blanks do not vary by more than a few percent through the different analyses and are orders of magnitude higher than for the sample (Sp). The BSP-ch display lower $^{235}U/^{238}U$, $^{230}Th/^{232}Th$ and $^{149}Sm/^{147}Sm$ values compared to the BSP (Fig. 6), showing that the chemistry protocol has an impact on the U, Th and Sm isotopes in solution. However, this effect remains insignificant compared to the Durango apatite U, Th, Sm contents, and does not influence the (U-Th)/He ages. Nevertheless, we always take care to have well characterized blanks, as, for natural apatite crystals, the U, Th and Sm contents are usually lower than for Durango.

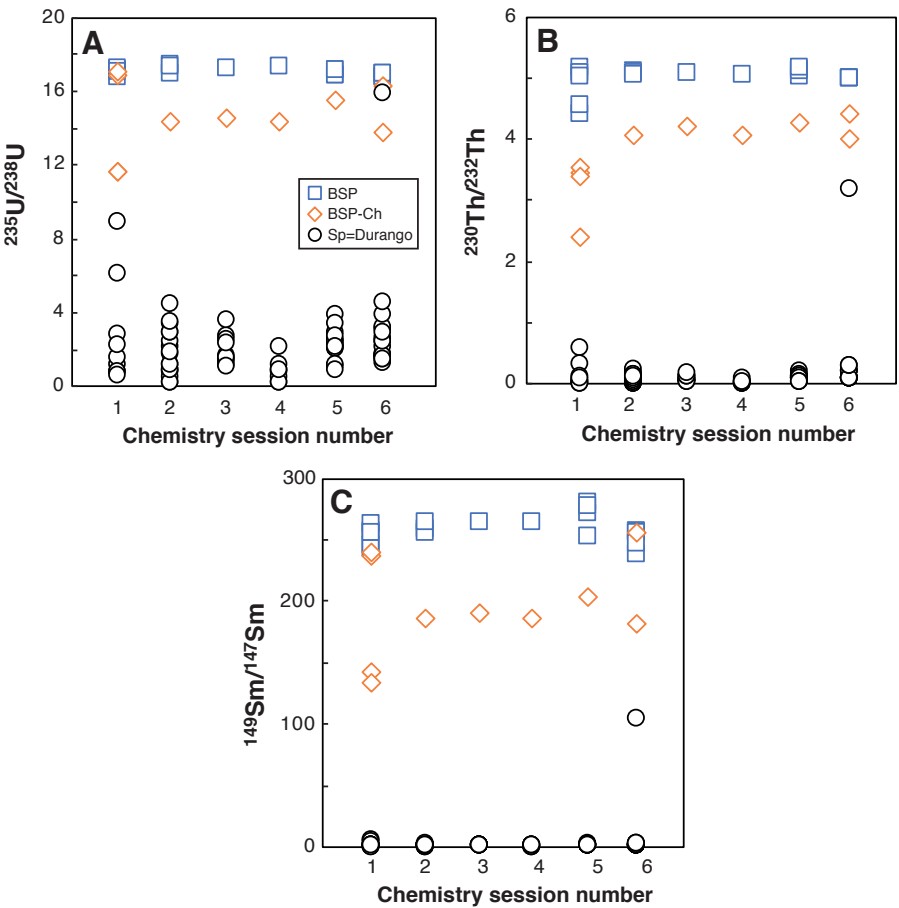

***Figure 6:*** *Evolution of the $^{235}U/^{238}U$, $^{230}Th/^{232}Th$ and $^{149}Sm/^{147}Sm$ ratios obtained for BSP, BSP-ch and Durango apatite Sp solutions, for 6 different chemistry sessions.*

In opposition to apatite, the chemical dissolution for zircon requires the acquisition and maintenance of more complex laboratory material, such as PFA vials, a high temperature/high pressure Parr® bomb and concentrated acids like HF. Tests

carried out on all types of blanks: BSP, BSP-ch and BSPch-Nb have been done over 18 series of sample analyses performed
between March 2016 and February 2018. Under constant vessel cleaning protocol and acid quality conditions, comparison of
the analyzed U and Th isotopes are presented in Figure 7. Isotopic ratios measured in BSP-ch ($^{235}U/^{238}U$=16.1±0.3;
$^{230}Th/^{232}Th$=4.8±0.5) showed a slight reduction of their values compared to those found in BSP ($^{235}U/^{238}U$=17.3±1.0;
$^{230}Th/^{232}Th$=5.1±0.2), indicating again a small contamination during the chemical protocol (Fig. 7). The loss of elements and
more specifically of Th when HF+HNO$_3$ acids are used has already been well described (e.g. Révillon and Hureau-Mazaudier,
2009; Yokoyama et al., 1999), and the effect can be seen on Figure 7, where the $^{235}U/^{238}U$ and $^{230}Th/^{232}Th$ ratios of BSP-ch are
lower than for BSP. In addition, the use of Nb capsules (BSP-ch-Nb) also impacts the U and Th budgets and leads to a massive
reduction of the ratios (maxima $^{235}U/^{238}U$=16.2±1.0; $^{230}Th/^{232}Th$=4.1±0.9 and recorded minima down to $^{235}U/^{238}U$=12.0;
$^{230}Th/^{232}Th$=1.7; Fig. 7).

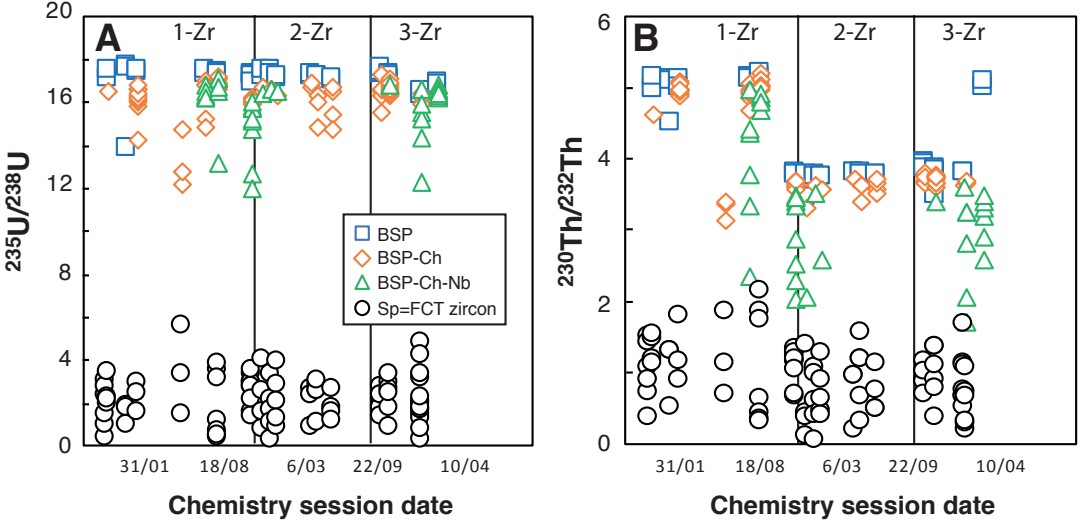

*Figure 7: Evolution of the $^{235}U/^{238}U$ and $^{230}Th/^{232}Th$ ratios obtained for BSP, BSP-ch, BSP-ch-Nb and FCT zircon solutions for
different chemistry session. 1-Zr, 2-Zr and 3-Zr refer to the spike solution names used during the dissolution.*

The impact of the niobium capsule on U and Th signals has already been noted by Reiners and Nicolescu (2007), and
reported to be more significant for the Th content. The differences in $^{235}U/^{238}U$ and $^{230}Th/^{232}Th$ ratios between the different
blanks (BSP, BSP-ch, BSP-Ch-Nb) are associated with some isotope fractionation, with a calculated decrease of respectively
4 to 6% for the $^{235}U/^{238}U$ ratio and 10 to 20% for the $^{230}Th/^{232}Th$ ratio, compared to the expected ratios given by the spiked

solutions without Nb. The shift for these ratios is systematic but variable from one solution to another, particularly for the $^{230}Th/^{232}Th$ ratio, showing values down to 60% of the expected spiked solution ratio. A correction of the impact of the niobium capsule on the U and Th solutions is considered as this effect can result in a shift of the zircon (U-Th)/He ages up to ~20%.

**3.4 Durango apatite and FCT zircon (U-Th)/He age reproducibility**

The Durango apatite is constantly analyzed in the laboratory to check for the He mass spectrometer sensitivity evolution though time as well as the evolution of (U-Th)/He age and U, Th and Sm contents. As the dissolution protocol for apatite has a very low impact on the U, Th and Sm content determination by ICP-MS analysis, the regular measurement of Durango apatite acts as a sensor and allows to detect any analytical problem. As an example, Figure 8 presents the values of the (U-
Th)/He ages, and Th/U and Sm/Th ratios acquired from March to December 2019 by both the Quad and VG lines, and values are reported in Table S1 (supplementary material). The mean of the (U-Th)/He age is 31.1±1.4 Ma in agreement with the age of 31.02±1.01 Ma from McDowell et al. (2005).

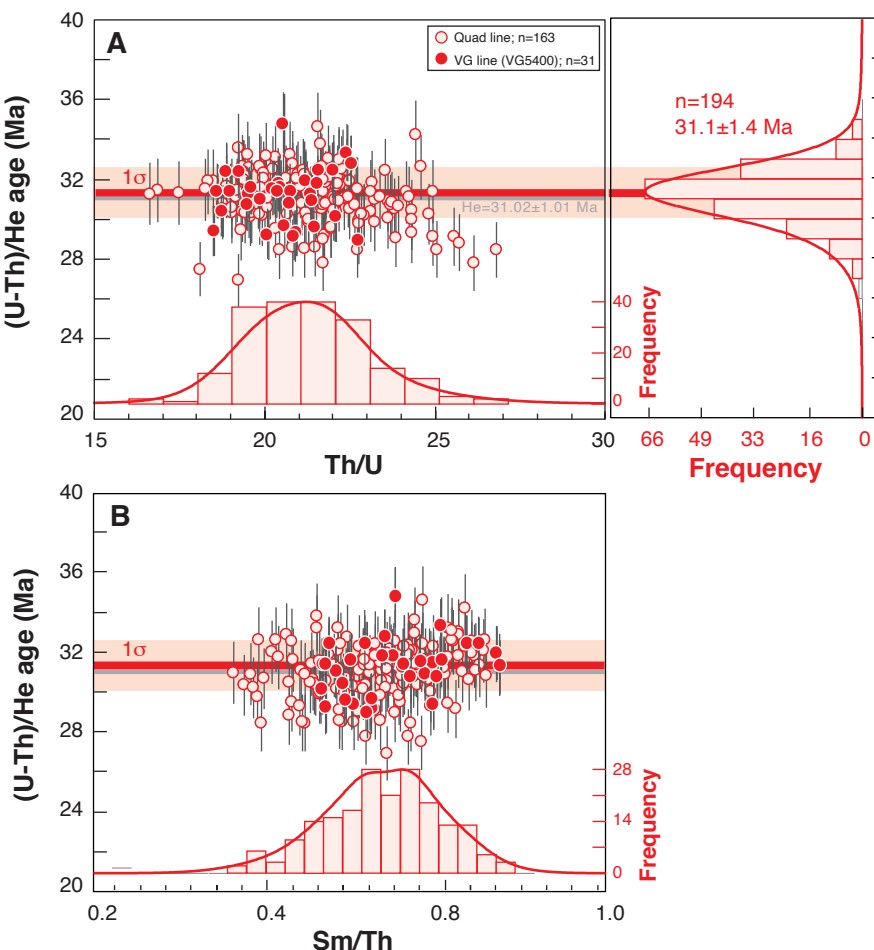

***Figure 8:*** *Durango (U-Th)/He age dispersion (1σ shown) as a function of Th/U (A) or Sm/Th ratios (B). Durango AHe ages and elemental ratios acquired from March to December 2019 using the Quad line (open red circles) or the VG line (filled red circles). Histogram representation of the He ages, Th/U and Sm/Th ratios were constructed using RadialPlotter (Vermeesch, 2009). Th/U and Sm/Th ratios have no unit but are expressed by mass (ng/ng).*

A typical mean error of <5% (1σ) is obtained on each AHe age by using either the quadrupole or magnetic sector mass spectrometers, without any evident difference over a large period of time. This error can be interpreted as the quadratic sum of the errors on the coupled analyses of the U-Th-Sm and the He contents and are associated with the calibration. Our (U-Th)/He ages on unknown apatites compare well within error with other laboratories (e.g., Ketcham et al., 2018). In addition, our strategy developed to determine the Ca concentration allows us to obtain the weight of the Durango fragment(s) and thus

to calculate the U, Th, Sm concentrations in ppm (Table S1). Mean values of U=19±4 ppm, Th=412±68 ppm and Sm=38±7

ppm have been obtained and the U content is similar to that obtained by Schneider et al. (2015) and Yanga et al. (2014).

Fish Canyon Tuff zircon crystals have been analyzed at the GEOPS laboratory as standards for the ZHe method. The U

and Th losses during dissolution, due to niobium impact, are corrected on the determination of the U and Th concentrations in

zircons. (U-Th)/He ages were obtained on 57 crystals of FCT zircons analyzed using the Quad and VG lines and are reported

in Figure 9 as a function of the Th/U ratio, and in Table S2 (supplementary material).

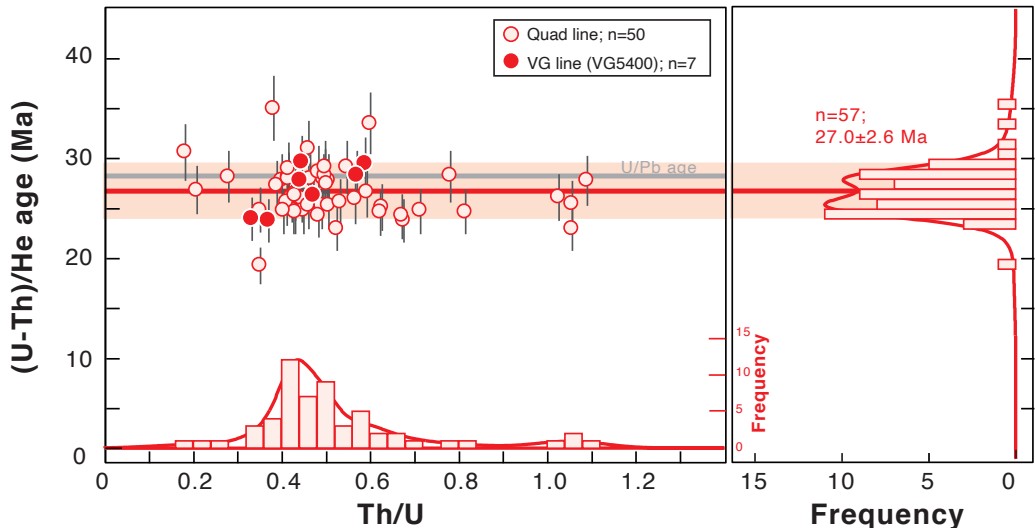


***Figure 9:*** *Fish Canyon Tuff zircon (U-Th)/He age dispersion (1σ shown) as a function of the Th/U ratio, for the data obtained in 2018 using the Quad and VG lines. Histogram representations of the (U-Th)/He ages and Th/U ratios were constructed using RadialPlotter (Vermeesch, 2009). An U/Pb age of 28.5±0.06 Ma has been published by Schmitz and Bowring (2001). Th/U is expressed by mass.*


We obtain a mean age of 27.0±2.6 Ma (1σ) and a mean Th/U ratio of 0.4 on a zircon crystal from the C.W. Naeser

collection (K/Ar age of 27.9±0.7 Ma; Naeser et al., 1991) (Fig. 9). The standard dispersion of the ZHe ages is ~9% and is

comparable to the natural dispersion observed in the ZHe values given in the literature (e.g., Ault et al., 2018; Guenthner et

al., 2014; Reiners, 2005). The Th/U dispersion of 37% also corresponds to the natural dispersion observed in the Th/U ratio of

the Fish Canyon zircon standard (e.g., Reiners et al., 2002). The (U-Th)/He age results are comparable with (U-Th)/He

literature data that range from 27.3±1.0 to 29.8±2.7 Ma (Dobson et al., 2008; Gleadow et al., 2015; Reiners et al., 2002; Tagami et al., 2003; Tibari et al., 2016). Th/U ratios vary between 0.42±0.15 (Tagami et al., 2003) and 0.63±0.14 Ma (Tibari et al., 2016) in the literature. The mean (U-Th)/He age obtained in this study is slightly younger, by 5%, than the U/Pb age of 28.5±0.06 Ma obtained by Schmitz and Bowring (2001), but still in good agreement within error bars (Fig. 9). The slight ZHe age difference could be explained by the variability of the measured Fish Canyon Tuff zircon ages as a function of sampling site (Gleadow et al., 2015). A second option is that, since similar ages are obtained by degassing either on the Quad or VG line, the slight shift in the (U-Th)/He age may be associated with the He content determination for only 1 or 2 percent, but is mainly associated to the U and Th content determination, and finally to the impact of the niobium precipitation during zircon dissolution. The loss of U, and more specifically of Th, associated to the use of HF+HNO$_3$ (Révillon and Hureau-Mazaudier, 2009; Yokoyama et al., 1999) and of Nb capsules (Reiners and Nicolescu, 2007) has already been taken into account in the blank correction. However, the slightly lower ZHe ages obtained in this study could also be associated with the impact of the zirconium brought into solution that could cause an additional slight loss in Th, not considered yet in the blank correction. To estimate the magnitude of this under-correction, additional work should be carried out to fully understand the U and Th isotopic and elemental fractionation during this chemical protocol.

## 4 Conclusion

This contribution presents the (U-Th)/He analysis protocols developed over the last ten years, at the GEOPS laboratory, Paris Saclay University, and shares all the empirical and analytical aspects observed during the different steps of the protocol: sample preparation, mineral hand picking, He analysis, mineral dissolution and U, Th and Sm content determination. In the light of our experience, we propose:

- a simple method to determine the temperature of the heated metallic (Pt and Nb) capsules that contain the apatite or zircon crystals during laser firing in the range 900-1200°C, using visible light emission wavelengths;
- a method to calibrate He sensitivity using quadrupole and magnetic sector mass spectrometers;
- the protocols to dissolve apatite and zircon crystals and to clean laboratory vessels after chemical digestion;
- the protocol to calibrate the U, Th and Sm spikes;
- the method used to track the U, Th and Sm blank evolution and determine U, Th and Sm contents.

We adopted the Durango apatite as a standard to perform He calibration and check for He, U-Th-Sm analytical problems, and we can thus determine (U-Th)/He ages with an error of less than 5% (1 $\sigma$). Our choice is also related to the fact that Durango is an easy-to-use mineral due to its high purity, its rapid dissolution protocol, and the strong reproducibility of its analyses. For the long-term quality control of the (U-Th)/He data, attention needs to be paid to evaluate precisely the drift of blanks through time, and that of the (U-Th)/He ages and Th/U ratios (with Sm/Th when possible) obtained on standards (Durango apatite and Fish Canyon Tuff zircon), especially when using quadrupole mass spectrometry.


**Code and data availability.** The Qt_LFT software and an Excel® file example is available in the Supplement, as well as Table S1 and Table S2 that present Durango and Fish Canyon Tuff zircon (U-Th)/He data. The data reduction Excel® workbook is available upon request to A.D.


**Supplement.** The supplement related to this article is available online at XXX

**Author contribution:** CG, RP, PS, LT, JM and DB designed the experiments, CG, RP, FA, AD, CS, FH, GM, GD participated in the data acquisition. LT developed the Monte Carlo simulation Qt_LFT software and AD the Excel® workbook automatization software. CG prepared the manuscript with contributions from all co-authors.


**Competing interests:** The authors declare that they have possible conflict of interest as C.G. is a member of the editorial board of the journal.

**Acknowledgments**

The analytical work and the He and VG lines building have been funded thanks to INSU-Relief, FORPRO and Tellus programs, INSU mi-lourd, Division de la recherche of the Paris Sud University, ERM Paris Sud program, ANR-06-JCJC-0079 and ANR-12-NS06-0005-01 HeDiff projects. M. Pagel is warmly thanked for his help in funding the cryogenic trap. IUT-Mesures-physiques internships trainees P. Marty, S. Lemaire, E. Cornier, M. Di Giacomo, G. Ya, P. Boutteville, M. Form, N. Etienne, C. Morelière, H. Vicente and B. Canguilhem are warmly thanked for their work and implication in LabView programming of the different parts of the Quad and VG lines. We thank P. Reiners and U. Chowdhury for sharing zircon fraction and their



knowledge on zircon dissolution protocols and J. R. Metcalf for the location of the FCT outcrop and advice. We warmly thank P.C. Hackspacher for the donation of gem quality Durango apatite crystals and C.W. Naeser for the donation of Fish Canyon Tuff zircon fraction. M. Moreira is thanked for his help on building the valve-controlling electronics and help on the building of the VG line. J.L. Birck is thanked for the loan of the external filament extinction pyrometer. P. Burckel, E. Douville and L.

Bordier are thanked for the U-Th-Sm analyses at IPGP and LSCE. Anonymous reviewers are thanked for their constructive reviews and the associated editor M. Tremblay and editor Greg Balco are thanked for review handling.

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

# Appendixes

**Appendix A:** U, Th, Sm and Ca spike calibration table (ppm and ppb are in $\mu$g/g and ng/g)

| Comment | Isotope | Volume | Solutions |
|---|---|---|---|
| **Spike solutions preparation** | | | |
| **MR:** concentrated mono-elemental mother solution and solid | $^{235}U_{MR}$ <br> $^{230}Th_{MR}$ <br> $^{149}Sm_{MR}$ <br> $^{42}Ca_{MR}$ | 10 ml <br> 5 ml <br> 100 ml | $HNO_3$ 5N 1 ppm (solution) <br> $HNO_3$ 5N 0.569 ppm (solution) <br> $HNO_3$ 5N 7 ppm (solution) <br> 2.44 mg $^{42}CaCO_3$ (solid) |
| **MR1:** first-dilution mother spike solution obtained by dilution from the concentrated MR mother solution or solid | $^{230}Th_{MR1}$=100 ppb <br> $^{42}Ca_{MR1}$=100 ppm | 10 ml <br> 10 ml | No need to dilute $^{235}U$ and $^{149}Sm$ (already at adapted concentration) <br> 1.8 ml $^{230}Th_{MR}$ + 8.2 ml $HNO_3$ 5N <br> 2.44 mg $^{42}CaCO_3$(solid) in 10 ml $HNO_3$ 5N |
| **MR2:** second-dilution mother spike solution obtained by mixing - dilution of the concentrated (MR) and first-dilution mother spike (MR1) | $^{235}U_{MR2}$=4 ppb <br> $^{230}Th_{MR2}$=4 ppb <br> $^{149}Sm_{MR2}$=4 ppb <br> $^{42}Ca_{MR2}$=1 ppm | 60 ml | 240 $\mu$L $^{235}U_{MR}$ (1 ppm) + <br> 2.4 ml $^{230}Th_{MR1}$ (100 ppb) + <br> 34 $\mu$L $^{149}Sm_{MR}$ (7 ppm) + <br> 598 $\mu$L $^{42}Ca_{MR1}$ (100 ppm) + <br> 56.727 mL $HNO_3$ 5N |
| **Spiked solutions for spikes calibration** | | | |
| **S:** concentrated standard solutions used for spike calibration | $^{238}U_S$ <br> $^{232}Th_S$ <br> $Sm_S$ natural <br> $Ca_S$ natural | 125 ml | $^{238}U$ tailored solution $HNO_3$ 5N 1015 ppm <br> $^{232}Th$ tailored solution $HNO_3$ 5N 993 ppm <br> Natural Sm solution $HNO_3$ 5N 1006 ppm ($^{147}Sm$ =14.99%; $^{149}Sm$ = 13.82%) <br> Natural Ca solution $HNO_3$ 5N 1003 ppm ($^{40}Ca$=96.94%, $^{42}Ca$=0.65%, $^{43}Ca$=0.14%) |
| **U$_s$(III):** freshly made multi-step dilutions to obtain 4 ppb $^{238}U$ mono-elemental standard solution | $^{238}U_S$ (I) <br><br> $^{238}U_S$ (II) <br><br> $^{238}U_S$ (III) | 8 ml <br><br> 8 ml <br><br> 8 ml | **I) 10 ppm $^{238}U$**=80 $\mu$L $^{238}U$ 1015 ppm + 7.92 ml $HNO_3$ 5N <br> **II) 100 ppb $^{238}U$**= 80 $\mu$L $^{238}U$ 10 ppm + 7.92 ml $HNO_3$ 5N <br> **III) 4 ppb $^{238}U$**= 320 $\mu$L $^{238}U$ 100 ppb + 7.68 ml $HNO_3$ 5N |

| | | | |
|---|---|---|---|
| **Th$_s$(III):** freshly made multi-step dilutions to obtain 4 ppb $^{232}$Th mono-elemental standard solution | $^{232}$Th$_S$ (I) | 8 ml | **I) 10 ppm $^{232}$Th** = 80 $\mu$L $^{232}$Th 993 ppm + 7.92 ml HNO$_3$ 5N |
| | $^{232}$Th$_S$ (II) | 8 ml | **II) 100 ppb $^{232}$Th** = 80 $\mu$L $^{232}$Th 10 ppm + 7.92 ml HNO$_3$ 5N |
| | $^{232}$Th$_S$ (III) | 8 ml | **III) 4 ppb $^{232}$Th** = 320 $\mu$L $^{232}$Th 100 ppb + 7.680 ml HNO$_3$ 5N |
| **Sm$_s$(III):** freshly made multi-step dilutions to obtain 4 ppb $^{149}$Sm mono-elemental standard solution | Sm$_S$ (I) | 10 ml | **I) 1 ppm Sm nat** = 10 $\mu$L Sm nat 1006 ppm+ 9.99 ml HNO$_3$ 5N |
| | Sm$_S$ (II) | 10 ml | **II) 70 ppb Sm nat** = 0.7 ml Sm nat 1 ppm + 9.3 ml HNO$_3$ 5N |
| | Sm$_S$ (III) | 8 ml | **III) 4 ppb Sm nat** = 0.46 ml Sm nat 70 ppb + 7.54 ml HNO$_3$ 5N |
| **SS:** freshly spiked standard solutions prepared using MR2 and S solutions | U$_{SS}$ | 2 ml | 50 $\mu$L $^{238}$U$_S$ (III) (4 ppb) + 50 $\mu$L $^{235}$U$_{MR2}$ (4 ppb) + 1.9 ml HNO$_3$ 1N |
| | Th$_{SS}$ | | 50 $\mu$L $^{232}$Th$_S$ (III) (4 ppb) $^+$ 50 $\mu$L $^{230}$Th$_{MR2}$ (4 ppb) + 1.9 ml HNO$_3$ 1N |
| | Sm$_{SS}$ | | 50 $\mu$L Sm$_S$ (III) (4ppb) + 50 $\mu$L $^{149}$Sm$_{MR2}$ (4ppb) + 1.9 ml HNO$_3$ 1N |
| | Ca$_{SS}$ | | 5 $\mu$L Ca$_S$ (1003 ppm) + 50 $\mu$L $^{42}$Ca$_{MR2}$ (1 ppm) + 1.945 ml HNO$_3$ 1N |

**Appendix B:** Equations used to determine the $^{238}$U, $^{232}$Th, $^{147}$Sm and $^{43}$Ca contents

The U, Th, Sm and Ca abundances are deduced from the measured $^{235}$U/$^{238}$U, $^{230}$Th/$^{232}$Th, $^{149}$Sm/$^{147}$Sm and $^{42}$Ca/$^{43}$Ca

10   isotopic ratios and the equations listed below (equations A, B, C, D and E) derived from Evans et al. (2005).

$$^{238}U = \left( m_{spk}(g) \times \left[ ^{235}U_{spk} \right] \times 10^{-9} \right) \times \frac{1}{\left( \left( \frac{^{235}U}{^{238}U} \right)_{spk} \right)} \times \left( \frac{\left( \frac{^{235}U}{^{238}U} \right)_{sp} - \left( \frac{^{235}U}{^{238}U} \right)_{BSP}}{\left( \frac{^{235}U}{^{238}U} \right)_{Nat} - \left( \frac{^{235}U}{^{238}U} \right)_{sp}} \right) \qquad (A)$$

$$^{232}Th = \left( m_{spk}\,(g) \times \left[ ^{230}Th_{spk} \right] \times 10^{-9} \right) \times \frac{1}{\left( \left( \dfrac{^{230}Th}{^{232}Th} \right)_{spk} \right)} \times \left( \frac{\left( \dfrac{^{230}Th}{^{232}Th} \right)_{BSP} - \left( \dfrac{^{230}Th}{^{232}Th} \right)_{sp}}{\left( \dfrac{^{230}Th}{^{232}Th} \right)_{sp}} \right) \tag{B}$$

$$^{147}Sm = \left( m_{spk}\,(g) \times \left[ ^{149}Sm_{spk} \right] \times 10^{-9} \right) \times \frac{1}{\left( \left( \dfrac{^{149}Sm}{^{147}Sm} \right)_{spk} \right)} \times \left( \frac{\left( \dfrac{^{149}Sm}{^{147}Sm} \right)_{sp} - \left( \dfrac{^{149}Sm}{^{147}Sm} \right)_{BSP}}{\left( \dfrac{^{149}Sm}{^{147}Sm} \right)_{Nat} - \left( \dfrac{^{149}Sm}{^{147}Sm} \right)_{sp}} \right) \tag{C}$$

$$^{43}Ca = \left( m_{spk}\,(g) \times \left[ ^{42}Ca_{spk} \right] \times 10^{-9} \right) \times \frac{1}{\left( \left( \dfrac{^{42}Ca}{^{43}Ca} \right)_{spk} \right)} \times \left( \frac{\left( \dfrac{^{42}Ca}{^{43}Ca} \right)_{sp} - \left( \dfrac{^{42}Ca}{^{43}Ca} \right)_{BSP}}{\left( \dfrac{^{42}Ca}{^{43}Ca} \right)_{Nat} - \left( \dfrac{^{42}Ca}{^{43}Ca} \right)_{sp}} \right) \tag{D}$$

with spk: spike, sp: sample, BSP: spiked blank, Nat: natural ratio, i.e., $^{235}U/^{238}U=0.00725268$; $^{149}Sm/^{147}Sm=0.92$ ($^{147}Sm=14.99\%$ and $^{149}Sm=13.8\%$) and $^{42}Ca/^{43}Ca=4.79$ ($^{42}Ca=0.647\%$ and $^{43}Ca=0.135\%$). We obtain abundances for each sample in nanogram ($10^{-9}$ g), with spike concentrations in ppb. In addition, the same equations can be reversed to determine the concentration of the spike isotopes $^{235}U$, $^{230}Th$, $^{149}Sm$ and $^{42}Ca$.

Guenthner et al. (2016) have already performed a complete work on determining apatite and zircon weights by measuring the Ca and Zr contents. Here, we obtain the weight of the apatite grain(s) from the measurement of $^{43}Ca$ (ng) and the composition of a pure fluoroapatite ($Ca_5(PO_4)_3F$) containing 40 wt. % Ca in one apatite crystal. Thus, the apatite weight is given by equation (E):

$$weight(\mu g) = \frac{^{43}Ca \times 10^{-9}}{0.135/100} \times \frac{1}{0.4 \times 10^{-6}} \tag{E}$$

The factor 0.135 refers to the natural isotope abundance of $^{43}Ca$.

**Appendix C:** Equations used for the (U-Th)/He age reduction

The (U-Th)/He age (in Ma) is calculated assuming a linear production of ⁴He with time, using the determined U, Th and Sm abundances and equations (F) and (G).

$$\frac{(U-Th)}{He}age(Ma) = \frac{^4He}{P^* \times 10^6}$$
(F)

where $P^*$ is the instantaneous production of ⁴He, i.e., the amount of ⁴He produced in one year, in ccSTP, and ⁴He is the measured ⁴He abundance in ccSTP. $P^*$ is calculated using the following equation:

35   $$P^* = \left(\frac{8 \times [^{238}U] \times 10^{-9}}{238} \times \lambda_{238_U} + \frac{7 \times [^{238}U] \times 10^{-9}}{235 \times 137.88} \times \lambda_{235_U} + \frac{6 \times [^{232}Th] \times 10^{-9}}{232} \times \lambda_{232_{Th}} + \frac{[^{147}Sm] \times 10^{-9}}{147} \times \lambda_{147_{Sm}}\right) \times 22414$$

(G)

with $\lambda_{238_U}(y^{-1}) = Ln(2)/(4.47 \times 10^9)$ ; $\lambda_{235_U}(y^{-1}) = Ln(2)/(7.04 \times 10^8)$ ; $\lambda_{232_{Th}}(y^{-1}) = Ln(2)/(1.40 \times 10^{10})$ ; $\lambda_{147_{Sm}}(y^{-1}) = Ln(2)/(1.06 \times 10^{11})$, and where [²³⁸U], [²³²Th] and [¹⁴⁷Sm] are the measured abundances in ng. The value of 137.88 is the ²³⁸U/²³⁵U natural isotopic ratio of uranium, and 1 mole occupies 22,414 cc at the standard pressure and temperature

40   conditions of 273.15 K and 1 atm (ccSTP). While ccSTP is not the SI unit for amount of substance, it is historically used by the noble gas community, and conversion to mol is easy using the conversion factor of 1 mol=22,414 ccSTP.