# Peer review of "Technical note: Analytical protocols and performance for apatite and zircon (U-Th)/He analysis on quadrupole and magnetic sector mass spectrometer systems between 2007 and 2020"

_Geochronology, 2021_

## Author Response (AR1)

Dear associated editor,

Thanks for your comments and suggestions that help improving the manuscript clarity.

As you suggested, we modified the title of the paper, and added in the abstract and introduction more information about the fact that this technical note presents the protocols for He thermochronometry developed at the GEOPS laboratory, Paris Saclay University. We corrected and answered in the paper to all suggestions, and transferred to the appendix section, table 3, and several equations in order to increase the readability of the text. We also added more details about the conversion between ccSTP unit to mol, as we agree that it is not a SI unit, but conveniently used by the community. In addition, two tables and the QTLFT software is available in the supplementary section.

Sincerely
Cécile Gautheron, on behalf of the co-authors

**Associate Editor Decision: Publish subject to minor revisions (further review by editor)** (07 Apr 2021) by Marissa Tremblay
Comments to the Author:
Dear Dr. Gautheron and coauthors,

Thank you very much for promptly responding to the two sets of reviewer comments we received following the discussion period of your technical note. In addition to the revisions you describe in your responses, I ask you to consider making the following changes to the technical note.

In both the title and the abstract, please clearly state the name of your laboratory facility. E.g., "Technical note: apatite and zircon (U-Th)/He analysis using quadrupole and magnetic sector mass spectrometry at the GEOPS Laboratory, Paris Saclay University." I view this as important information to have up front, because the reality is that different labs make (U-Th)/He analyses differently, and labs other than your own will have different procedures for certain parts of the measurement process that are not detailed here.

Thanks for your advice, we modified the title to be "" and also state clearly in the ms that the protocol is the one developed at the GEOPS Laboratory, Paris Saclay University. Also in the text, we add a sentence to explain that this technical note is only reporting the protocol we developed, but it is a protocol among many others.

Line 41: Presumably the magnetic separation takes place before the heavy liquids separation? If so, I would reorder the text in this sentence to reflect the sequence of events in the separation procedure.

We are doing the same sequence, with heavy liquids before magnetic separation.

Line 50: Is the modified QTLFT software and excel file used to calculate Ft corrections publicly available? If so, where can the reader find this? And if not, would you be willing to make this available as a supplement to your technical note?

Yes, the modified QTLFT is available, and we add it as a supplement of the technical note

Line 83 and elsewhere: I had to look up the meaning of inox. I recommend either changing this to stainless steel, or putting stainless steel in parentheses after the word inox.

We did it, as it was already mention by one reviewer

Line 87: Change 'Cupper' to 'Copper.'

Done

Equation 2: There is an extra multiplication sign after the equals sing.

We corrected it

Line 160: Change analyzes to analyses.

Done

Line 167: What are the units for 'He' here in parentheses?

We add the information (cps)

Sample digestion protocol: It is a bit confusing to me that the spike concentrations are listed as approximate values in this section. Once I read the next section on spike calibration, this makes sense, so I recommend moving the spike calibration section above the sample digestion protocol section. Also, what does ppb mean here? Is it ng/g or ng/mL? Please specify here and elsewhere (i.e., where ppm is used).

Effectively, it is clearer like that.
We added also the significance of the meaning of ppm and ppb that are in are in $\mu$g/g and ng/g

Table 3: I recommend moving this to be supplementary material. The details are highly specific to the spike solutions used in this particular lab.

Ok, we moved Table 3 in the supplementary material

Lines 257-259: Is this Excel worksheet publicly available? If so, where? And if not, would you consider making it accessible as supplementary material to your technical note?

The excel worksheet is available upon request only, as it is better that user contact directly Alexis Dercyke.

Lines 288-289: I appreciate the inclusion of the conversion factor you are using to go from ccSTP to moles, as this is often left out in the literature. However, this begs the question of why the authors do not report helium amounts in units of moles in the first place? I strongly encourage the authors to simply use moles here and throughout the manuscript and supplement. ccSTP is not an SI unit, and moles is the appropriate unit to use for the (U-Th)/He age calculation anyway.

The use of ccSTP or cc is historical in the noble gas community and most of the (U-Th)/He scientists are used to this unit. In understand that it is not a SI unit and so we add a sentence about this in the ms and explain how to make the conversion.

Figure 5: I think using a log-log plot here is a bit deceptive, because large magnitude deviations from the line are hard to see. I recommend adding a subplot that demonstrates the magnitude of the deviation from the linear fit for each of the points.

We have now two subplots in Fig 5, where Fig. 5B has a linear scale that better shows the deviations. Figure 5B shows the dispersion and we discuss it in terms of difficulties in analysing U and Th in extremely small grains.

---

## Author Response (AR2)

Pr. Cécile Gautheron
UMR Géosciences Paris Saclay
Université Paris Saclay
91405 Orsay, France
+33 1 69 15 67 83
cecile.gautheron@universite-paris-saclay.fr

[Figure]

FACULTÉ
DES SCIENCES
D'ORSAY

Orsay, the 28th of April 2021

Dear associate Marissa Tremblay and Greg Balco,

Thanks for the final comments and the editorial handling of this technical note.

In the final version, we made all the asked corrections, and gave the precision in the abstract but also added accuracy. Effectively, we are using Durango apatite to access the mass spectrometer sensitivity but also detect changes in instrument sensitivity and potential analytical issues. We have also adapted the text regarding the ccSTP unit and added the equivalent values in unites of moles.

Sincerely
Cécile Gautheron, on behalf of the co-authors